# Modification of PET Ion-Track Membranes by Silica Nanoparticles for Direct Contact Membrane Distillation of Salt Solutions

**DOI:** 10.3390/membranes10110322

**Published:** 2020-10-30

**Authors:** Ilya V. Korolkov, Azhar Kuandykova, Arman B. Yeszhanov, Olgun Güven, Yevgeniy G. Gorin, Maxim V. Zdorovets

**Affiliations:** 1L.N.Gumilyov Eurasian National University, Satpaev str., Nur-Sultan 010008, Kazakhstan; k.azhar823@gmail.com (A.K.); arman_e7@mail.ru (A.B.Y.); gorineg@mail.ru (Y.G.G.); mzdorovets@gmail.com (M.V.Z.); 2The Institute of Nuclear Physics, Ibragimov str., Almaty 050032, Kazakhstan; 3Department of Chemistry, Hacettepe University, Beytepe, 06800 Ankara, Turkey; guven@hacettepe.edu.tr; 4Ural Federal University, Mira str. 19, 620002 Ekaterinburg, Russia

**Keywords:** desalination, hydrophobic modification, track-etched membranes, membrane distillation, silane, silica nanoparticles

## Abstract

The paper describes desalination by membrane distillation (MD) using ion-track membranes. Poly(ethylene terephthalate) (PET) ion-track membranes were hydrophobized by the immobilization of hydrophobic vinyl-silica nanoparticles (Si NPs). Si NPs were synthesized by the sol-gel method, and the addition of the surfactant led to the formation of NPs with average size of 40 nm. The thermal initiator fixed to the surface of membranes allowed attachment of triethoxyvinyl silane Si NPs at the membrane surface. To further increase hydrophobicity, ethoxy groups were fluorinated. The morphology and chemical structure of prepared membranes were characterized by SEM, FTIR, XPS spectroscopy, and a gas permeability test. Hydrophobic properties were evaluated by contact angle (CA) and liquid entry pressure (LEP) measurements. Membranes with CA 125–143° were tested in direct contact membrane distillation (DCMD) of 30 g/L saline solution. Membranes showed water fluxes from 2.2 to 15.4 kg/(m^2^·h) with salt rejection values of 93–99%.

## 1. Introduction

The search for effective and affordable water treatment methods is an urgent task due to a decrease in drinking water reserves, and an increase in population and industrialization [1]. The United Nations reported that around 700 million people currently live in water-scarce countries. It is expected that, by the year 2025, that 1.8 billion people will live in regions with water scarcity [2]. Among water pollutants, the most important ones are inorganic salts, oil products, surfactants, pesticides, and phenols. The most effective methods of water treatment is membrane technologies including osmosis, filtration and membrane distillation (MD) [3].

MD was considered as one of the most promising methods due to the following advantages, such as mild operating conditions, high rates of rejection, low working pressure, less sensitivity to fouling, and low feed temperature requirements [4,5,6,7,8]. MD can be realized in five types of configurations: air-gap MD, vacuum MD, sweeping gas MD, permeate gap MD, and direct contact MD (DCMD). Despite some limitations, the most studied type is DCMD due to its simplicity and availability [3,9,10,11,12,13,14]. DCMD can be successfully used to purify water from salts [15,16,17,18,19], heavy metals [20,21,22], dyes [23,24], radioactive wastes [25,26,27,28], acid solutions [29], urea [30], mining, and industrial wastewater [31,32].

MD is a thermally driven process where membrane serves as a barrier for liquid permeation, but it is a medium for vapor transfer. Thus, membranes used in MD should be hydrophobic to prevent fluid leak [33] and the efficiency of the MD process mostly depends on the characteristics of the membrane used. Commercially available hollow fiber and flat sheet polymer membrane based on poly(tetrafluoroethylene), poly(vinylidene fluoride), and polypropylene are typically used in MD [34,35,36,37,38]. However, wetting, fouling, and high thermal energy demand of such kind of membranes are the main drawbacks for successful application in MD [8,39,40]. To overcome these drawbacks, other types of membranes can be used [11,41,42,43,44,45], for instance, non-fiber, multilayered, composite membranes, ion-track membranes and methods of modification such as plasma, thermal, laser treatment, layer-by-layer assembly, cross-linking, coating, and graft polymerization of hydrophobic compounds [46,47,48,49,50]. The method of modifying membranes using silica nanoparticles has also been recommended as a method for producing superhydrophobic or omniphobic membranes. However, to date, there are only a few works devoted to using modified flat sheet membranes in MD [51,52,53]. The development of a reliable method of hydrophobic silica nanoparticles immobilization to prepare membranes suitable for MD is a challenging task.

In this article, we consider the covalent attachment of silica nanoparticles carrying C=C bond on their surface by thermo-induced grafting on pre-modified PET track-etched membranes (TeMs) with azobis(2-amidinopropane) dihydrochloride (ABAP) and subsequent modification with fluorine-containing silane. Ion-track membranes have regular geometry of the pores with the ability to control their amount per unit area and excellent distribution of pore sizes together with small thickness and low tortuosity [54]. Their unique properties can lead to enhancement of selectivity and efficiency of MD process and make them suitable for potential application in MD [27,28,55]. Researches on the immobilization of silica nanoparticles on ion-track membranes for use in MD, which would contribute to reduce membranes fouling and wetting have not been elaborated yet.

## 2. Materials and Methods

### 2.1. Chemicals

Triethoxyvinylsilane (TEVS), 2,2′-azobis (2-methylpropion-amidine) hydrochloride (ABAP), N-(3-dimethylaminopropyl) -N′-ethylcarbodiimide (EDC), pentafluorophenol (PFP), 1H, 1H, 2H, 2H-perfluorodecyl triethoxysilane (PFDTS), and isopropanol were supplied by Sigma Aldrich (Hong Kong, China). All other chemicals and solvents such as o-xylene, sodium chloride, sodium lauryl sulfate, aqueous ammonia, ethanol, acetic acid, sodium hydroxide, and methanol had purity of analytical grade. The reagents were purified before use by recrystallization, distillation, or column chromatography. In all experiments, deionized water (18.2 MΩ) obtained from Aquilon-D301 (Aquilon, Moscow, Russia) was used.

### 2.2. Synthesis of the Silica Nanoparticles

The synthesis of Si NPs with C=C bonds was based on the sol-gel method described in [56]. To reduce the average NPs size, the addition of surfactants was investigated. 0.0005 M–0.006 M of sodium lauryl sulfate was dissolved in 30 mL of deionized water. Then, 3 mL of triethoxyvinylsilane was added dropwise with vigorous magnetic stirring for 1 h. After that, 3 mL of ammonium hydroxide (25%) was added dropwise using a dropping funnel, continuing vigorous magnetic stirring. The resulting solution can be transparent or turbid depending on surfactant concentration with alkaline pH. If the resulting solution is transparent, acetic acid was added to adjust the pH to neutral, leading to the precipitation of nanosized Si NPs with surfaces modified with vinyl groups. The particles were separated from the mixture by centrifugation (6000 rpm), washed several times with water. Then NPs were dispersed in ethanol. The resulting solution of silica nanoparticles in ethanol was subsequently used to modify PET ion-track membranes.

### 2.3. Fabrication of Ion-Track Membranes and Their Modification by Silica NPs

PET ion-track membranes were prepared by irradiation of PET film (12 µm thickness) with Kr ions on accelerator DC-60 (Institute of Nuclear Physics, Nur-Sultan, Kazakhstan) with an energy of 1.75 MeV/nucleon and ion fluence of 1 × 10^8^ ion/cm^2^. Then membranes were obtained by photosensitization for 30 min from each side and chemical treatment in 2.2 M NaOH at 85 °C for 140 s, 165 s, 180 s, and 195 s to prepare membranes with pore sizes of 247, 300, 350, and 410 nm. Chemical treatment of PET film also led to the hydrolysis of the ester groups of PET with breakage of the polymer backbone and formation of –COOH and –OH groups at the chain termini [57,58]. The modification of PET ion-track membranes was performed according to the scheme presented in Figure 1.

At the first stage, 2,2′-azobis (2-methylpropion-amidine) hydrochloride (ABAP) was covalently bonded with COOH-end groups of PET [59]. PET ion-track membranes were kept in an ethanol solution of 0.2 M pentafluorophenol (PFP) and 0.1 M N-(3-dimethylaminopropyl) -N′-ethylcarbodiimide (EDC) for 2.5 h. Then, the membranes were kept for 20 h in a 20% aqueous-alcoholic solution (50% vol.) of ABAP and sequentially washed in tetrahydrofuran, dried with argon and immediately used for further grafting of Si NPs due to the instability of the resulting compound (PET-ABAP ion-track membranes were prepared).

At the second stage, the ethanol solution of prepared Si NPs was passed through the membranes from both sides using vacuum pump to fill the nanochannels with Si NPs. Membranes with Si NPs were immersed in ethanol solution of 0.2% ABAP. The solution was purged with argon and kept at 75 °C for 3 h to initiate grafting of Si NPs to the surface of PET ion-track membranes. After the reaction, membranes were washed in ethanol to flush away non-grafted NPs and dried (PET-Si ion-track membranes were prepared).

At the third stage, prepared membranes were modified with 1H,1H,2H,2H-perfluorodecyl triethoxysilane (PFDTS). PFDTS in the concentration range from 1 to 20 mM were prepared in o-xylene, membranes were kept in this solution for 1–24 h. After the reaction, membranes were washed in pure o-xylene to clean the surface from non-reacted PFDTS (PET-Si-F ion-track membranes were prepared).

### 2.4. Methods of Characterization

FTIR spectra were recorded on Agilent Cary 600 spectrometer with ATR accessory (Agilent Technologies, Mulgrave, Australis) in scan range of 400 to 4000 cm^−1^ and resolution of 0.24 cm^−1^ to evaluate chemical changes at each stage of membrane modification. XPS were recorded on a Thermo Scientific K-Alpha spectrometer (the Ural Center for Shared Use “Modern Nanotechnology” Yekaterinburg, Russia) at pressure of 2 × 10^−6^ Pa or lower in the analysis chamber. Processing of the data was carried out by Avantage software. Morphology and pore size of the membranes were characterized by using the scanning electron microscope (SEM) JEOL JSM-7500F (JEOL Ltd, Akishima, Tokyo, Japan). Pore size of the membranes was evaluated by gas flow rate measurement at a pressure drop of 20 kPa [60]. The hydrophobicity of the membranes were characterized by contact angle (CA) and liquid entry pressure (LEP) measurements.

CA of water was measured from five different positions of the sample using Digital Microscope 1000× magnification (OEM, Ningbo, China) using static drop method. LEP was determined according to recommendations described in [61,62,63]. A circular sample with the radius of 1.25 cm was clenched inside the sealed chamber, and a test was run with air at gradually increasing pressure. A pipette with a 0.7-mm diameter capillary was used for LEP evaluation.

### 2.5. Direct Contact Membrane Distillation

Direct contact membrane distillation (DCMD) was used to determine separation performance of hydrophobized PET TeMs. DCMD is schematically presented in Figure 2. This system consists of four thermocouples type-T (T1,T2,T3,T4). The membrane was placed in a cell for MD process, the flow rate on permeate and feed side was controlled and kept constant at 227 ± 3 mL/min and 453 ± 3 mL/min respectively using Easy load Cole-Parmer Masterflex L/s 77200-62 (Cole-Parmer Instrument Co, Vernon Hills, IL, USA). The temperature difference was kept at 70 °C. The permeate flux was measured by weighing the mass of liquid collected on permeate side at fixed time intervals (30 s) on a balance (±0.01 g). Then, solution of NaCl with concentration of 30 g/L was chosen as feed, since it is the average salinity of seawater. The efficiency of salt rejection was evaluated conductometrically using a Hanna Instruments conductometer HI2030-01 (HANNA Instruments, Cluj, Romania).

The flow rate was calculated by the formula:(1)$$J=\frac{\Delta m}{A\Delta t}$$
where *J* denotes water flux (g/m^2^·h) and Δ*m* the difference of mass in permeate side (g) per unit time Δ*t* (h) and effective area of membrane *A* (m^2^).

Degree of salt rejection (R) was calculated by the equation:(2)$$R=100-\;\left(\frac{C_{real}}{C_{fic}}\cdot 100\%\right)$$
(3)$$C_{real}=\;\frac{\Delta \sigma \cdot 1000}{2.3}$$
(4)$$C_{fic}=\;\frac{\Delta m\cdot C_{feed}}{m_{p}}$$
where *R* denotes the degree of salt rejection %, *C_real_* the concentration of NaCl in permeate side after MD, g/L, calculated according conductivity (conductivity of 1 mg/L NaCl solution is 2.3 µS/cm); *C_fic_* is the theoretical concentration of NaCl (providing that feed solution passed without purification), g/L; Δ*σ* is the difference in conductivity of permeate solution before and after MD, µS/cm; 2300 µS/cm is the change in the conductivity of the solution with the addition of 1 g/L of NaCl; Δ*m* is the permeate gain after MD, g; *C_feed_* is the initial concentration of salt in feed solution, g/L, and m_p_ is the mass of water from the permeate side before MD, g.

## 3. Results

### 3.1. Synthesis of Silica Nanoparticles

The synthesis of Si NPs was based on the procedure described in the work [56]. To reduce NPs size, surfactant was used. SEM analysis was used to estimate average size of NPs. Before SEM analysis, silica NPs were immobilized on the surface of ion-track membranes by filtration of colloid solutions. Figure 3 shows SEM images of prepared NPs with addition of sodium lauryl sulfate in the reaction (0.0005 M, 0.0008 M, 0.0015 M, 0.002 M, 0.003 M, and 0.006 M).

SEM images show that the higher the concentration of surfactants, the smaller the size of the nanoparticles. At the same time, at surfactant concentration of 0.0005 M, 0.0008 M, 0.0015 M and 0.002 M the formation of large particles along with small ones can be noticed thus a large dispersion of NPs in size is observed. At concentration of 0.003 M, average size of NPs is 40 ± 4 nm, increasing surfactant concentration to 0.006 M led to formation of NPs with average size of 23 ± 3 nm. It is also seen that at this concentration, the morphology of membrane surface only slightly changed. Table 1 shows that CA of PET TeMs depends on concentration of surfactant. With an increase in the concentration of surfactants, a decrease in CA occurred. It can be associated with a less rough surface morphology of the resulting membrane. Also, a decrease in CA can be due to the surfactant remaining on the surface of the NPs at higher surfactant concentrations (0.006 M). Thus, 0.003 M of surfactant was found to be the optimal concentration considering the size of NPs and CA for preparation of PET TeMs. NPs prepared in this way were further used to fix them on PET ion-track membranes.

### 3.2. Preparation of Hydrophobic Membrane

Hydrophobization of PET TeMs was carried out according to the scheme presented in Figure 1. First of all thermal initiator ABAP was attached to the surface via reaction with carboxylic-end groups of PET membranes [59]. Then, the ethanol solution of synthesized Si NPs (40 nm) were pumped through the membrane pores. Vinyl-Si NPs were grafted to the PET ion-track membranes by the reaction of vinyl groups of vinyl-Si NPs via radical sites generated by thermal decomposition of ABAP on the PET surface. Covalent bonding of vinyl-Si NPs on PET surface will prevent the washing off of NPs during exploitation. Finally, to increase hydrophobic properties of prepared membranes, they were covered with fluorinated silane PFDTS via hydrolysis. The effect of PFDTS concentration and time of the reaction on CA were studied. Results are presented in Figure 4. Optimal conditions to achieve the highest value of CA are 20 mM of PFDTS with a reaction time of 24 h.

Every step of chemical modification was controlled by methods of gas-permeability, CA, LEP analysis, FTIR and XPS spectroscopy. Membranes with initial pore diameters of 247, 300, 350, and 410 nm were hydrophobized. Values of CA, LEP, pore diameter changes, and other parameters are summarized in Table 2. The results of CA measurements from 3 different positions for each sample of PET TeMs are presented in Figure 5.

Increasing of grafting degree led to increasing of CA from 49° for untreated PET TeMs to 143° for PET-Si-F membranes. Contact angle depends on both chemical natural and roughness of the surface [64,65]. The method applied in this work allows changing of both parameters.

Analysis of previously published works on the use of silica NPs for hydrophobization of membranes shows us that CA of PVDF flat sheet and micropillared membranes can be increased up to 150°–160° [51,52] and to 150° for PLA fabrics [66].

LEP was estimated experimentally and using model of combination of computational fluid dynamics and genetic programming (CFD-GP) developed by Chamani et al. [63]. A good correlation was found between the experimental data and CFD-GP model. The experimental setup used for experimental LEP analysis does not allow to increase the pressure more than 4.3 bar, hence for some samples the experimental LEP exceeded the maximum value of pressure. According to CA and LEP data, all membranes with different pore diameters can be used in MD in connection with recommendations [33]. Also, it should be noted that at a pore density of 1 × 10^8^ pore/cm^2^, membranes with pore diameters more than 400 nm lead to pore overlapping and a decrease in strength [67]. Therefore, membranes with larger pore diameters could not be used.

The stability of hydrophobized layer was investigated by holding the modified membranes in water at 75 °C which is comparable to the conditions of DCMD. Minor changes in CA (Figure 6 and no changes in LEP were detected. A slight decrease in the contact angle may have occurred due to the washout of a small amount of unattached nanoparticles or 1H,1H,2H,2H-perfluorodecyl triethoxysilane.

Changes in chemical structure during modification were controlled by FTIR spectroscopy using ATR-accessory. The main absorption bands for the initial PET ion-track membranes (Figure 7) are determined as 2970 cm^−1^ (aromatic CH), 2912 cm^−1^ (aliphatic CH), 1713 cm^−1^ (C=O group), 1615, 1470, 1430, 1409 cm^−1^ (aromatic vibrations of the carbon skeleton), 1340 cm^−1^ (OCH bending), stretching vibrations of C(O)-O bonds of ether groups (1238 cm^−1^), 1096 cm^−1^ (C-O stretching), 1017 cm^−1^ (ring CCC bending) and 970 cm^−1^ (O–CH_2_) [45,68]. After the modification with ABAP, no significant changes were observed in FTIR spectra. After immobilization of Si NPs on PET ion-track membranes, new peaks appeared at 1080 and 1132 cm^−1^ (Si-O-Si stretching vibration), and 1602 cm^−1^ (C=C stretching vibration) [69]. Upon grafting of Si NPs on the surface, peak at 1602 cm^−1^ decreased, which is seen from Figure 7b.

It indicates reaction of C=C bonds as graft polymerization proceeds. For a quantitative assessment, the values of the I_1602_/I_1410_ band ratio indexes were calculated on the basis of respective peak intensity (I). Band ratio index I_1602_/I_1410_ for the PET TeMs—Si is 2.88 while for the samples PET TeMs-Si grafted is 1.43.

Successful surface modification with PFDTS was proved by the appearance of peaks at 1150 and 1204 cm^−1^ ascribed to CF_2_ and CF_3_ symmetric stretching [52].

Surface chemical structure and elemental analysis of the membranes were also studied by more sensitive method XPS. XPS can analyze chemical composition of top layer (~10 nm) of the membranes. Elemental composition obtained from survey XPS spectra (not shown) of PET ion-track membranes at each stage of modification is presented in Table 3. Initial PET ion-track membrane is composed of 72.5% C and 27.5% O, modification with ABAP led to appearance of 2.6% N. PET-Si-F membranes were found to have 4.1% of Si and 2.5% of F. Typical high-resolution XPS spectra are presented in Figure 8. High-resolution Si2p spectra (Figure 8c) consists of one peak at 102.5 eV related to Si-O-Si, and F1S (Figure 8d) is also consisting of single peak related to C-F. The C1s peaks of original PET ion-track membranes are attributed to C-C/C-H at 284.6 eV, C-O/C-OH at 286.5 eV, and C=O at 288.7 eV [70,71,72]. Coating with PFDTS led to appearance of two additional peaks at 291.4 eV and 293.9 eV correspond to CF_2_ (1.6%) and CF_3_ (0.3%) respectively. The ratio of intensities of CF_2_ and CF_3_ peaks in XPS spectra is close to atomic ratio depicted in the molecule of PFDTS. The results of XPS spectra are consistent with previously published works related to silane and fluorine-containing silane coating of different materials [73,74].

Figure 9 shows cross-section view of initial and modified PET ion-track membranes. It is seen that Si NPs are located inside the channels. However, they are more concentrated on the surface and pore mouths. It can be due to difficulty of adsorption of Si NPs on inner pore walls since they are perpendicular to the flow trajectory.

### 3.3. Membrane Distillation

Membrane distillation by using hydrophobic PET ion-track membranes prepared at optimal conditions with different pore sizes of 201, 263, and 315 nm was performed in DCMD. Figure 10a shows the effect of pore diameter on average water flux.

The average water flux for membranes with effective pore diameter (measured by gas permeability) of 210 nm is 2.2 kg/(m^2^·h), 263 nm; 6.5 kg/(m^2^·h) and 315 nm; 15.4 kg/(m^2^·h). The increase in water fluxed is directly associated with increase in membrane porosity. Degree of salt rejection was evaluated by measuring electrical conductivity [45] of permeate side (Figure 10b). It is very well known that conductivity is sensitive to the slightest change in salt concentration [75,76]. We observed slight increase in conductivity with time for membranes with pore diameters of 201 and 263 nm and more intense increase for membranes with pore diameter of 315 nm. Degree of salt rejection was found to be 99%, 98%, and 93% for hydrophobized membrane with pore diameters of 201, 263, and 315 nm respectively. It should be noted that unmodified PET ion-track membranes were also tested in MD, however the degree of salt rejection measured by electroconductivity was close to zero. Probably filtration is occurred instead of MD, since unmodified membranes have unsatisfied properties to use them in MD.

Figure 11a shows SEM images of dried PET-Si-F membranes after DCMD of saline solution without washing. A large amount of contamination is present on the membrane surface. After washing in warm water for 8 h, most of the contaminations were washed away. However, the Si NPs were retained on the membrane surface. CA of the membranes slightly changed from 132° ± 5° to 127° ± 6°.

The results show the possibility of using silica NPs for hydrophobization of PET ion-track membranes and application in water purification by DCMD. Compared with the previously obtained results on the hydrophobization of PET ion-track membranes [11,45,55], the proposed method allows to reach a higher permeability with an appropriate salt rejection degree. To compare results of water purification by MD using other types of membranes [3,16,23,77,78,79], the use of PET ion-track membranes leads to relatively low permeate flux, which is primarily due to the low porosity of these membranes. At the same time, accurate pore sizes and the narrow pore size distribution of PET ion-track membranes lead to the accurate separation of different liquids.

## 4. Conclusions

In this study, we have shown that the application of the method of immobilization of hydrophobic silica nanoparticles onto PET ion-track membranes allowed significant change its hydrophobic properties. CA increased from 49 to 143° for membranes with pore diameter of 152 nm, to 135°, 132°, and 125° for membranes with diameters of 201, 263, and 315 nm, respectively. PET ion-track membranes with a large initial pore diameter (410 nm) were hydrophobized that allowed reaching LEP to 3.5 bar. SEM analysis showed changes in morphology of the surface after modification, XPS and FTIR spectroscopy confirmed success modification of the membranes. DCMD test showed maximum flux of 15.4 kg/(m^2^·h) with degree of salt rejection of 93%, 6.5 kg/(m^2^·h) with 98%, and 2.2 kg/(m^2^·h) with 99%. The results showed a high potential for the possibility of using the obtained membranes in membrane distillation.

## Figures and Tables

**Figure 1 membranes-10-00322-f001:**
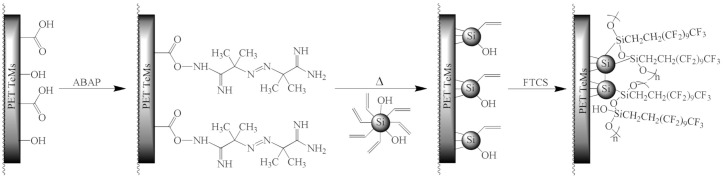
Scheme of PET ion-track membrane modification.

**Figure 2 membranes-10-00322-f002:**
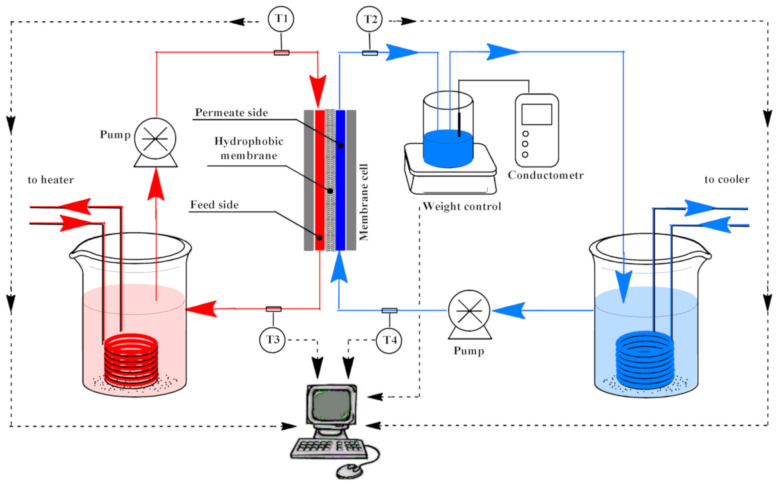
Scheme of membrane distillation.

**Figure 3 membranes-10-00322-f003:**
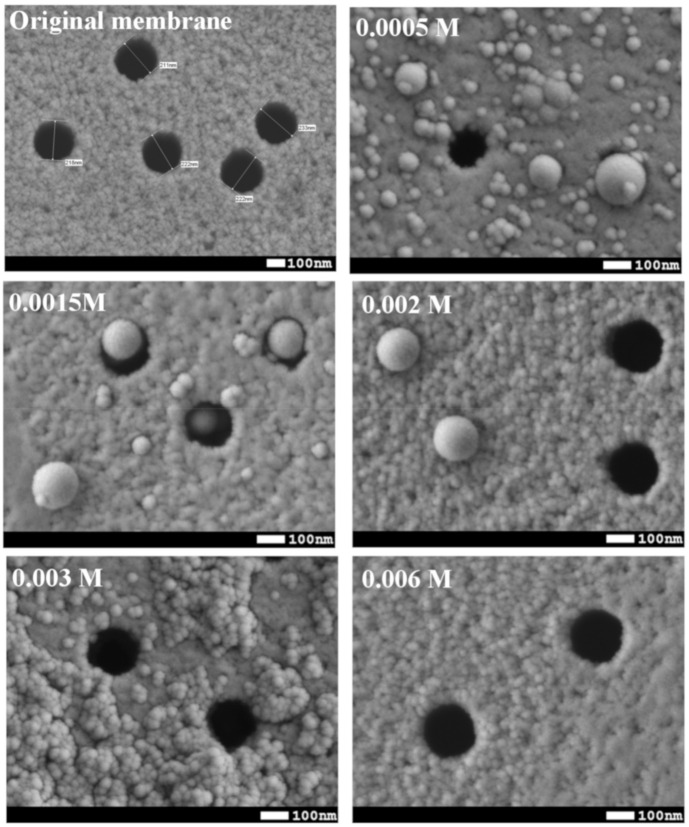
SEM images of original PET membrane and PET-Si NPs membranes at different concentration of surfactant.

**Figure 4 membranes-10-00322-f004:**
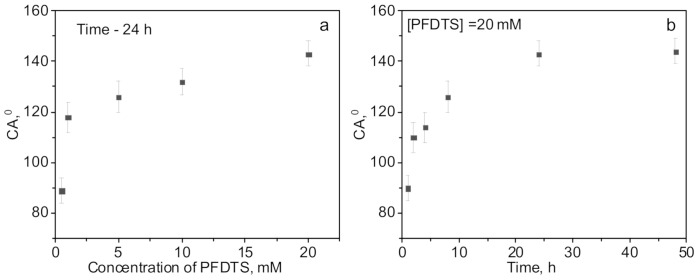
CA of PET-Si-F membranes at different reaction time (**a**) and concentrations of PFDTS (**b**) (the average pore size of the original membrane is 247 nm).

**Figure 5 membranes-10-00322-f005:**
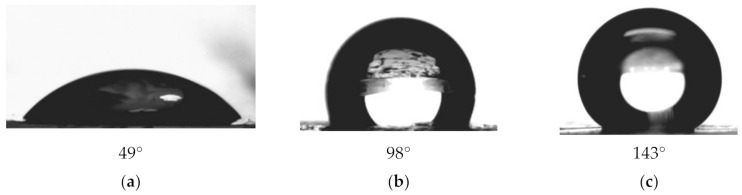
CA for original membrane (d ≈ 247 nm) (**a**), PET-Si membranes (**b**) and PET-Si-F membranes (**c**).

**Figure 6 membranes-10-00322-f006:**
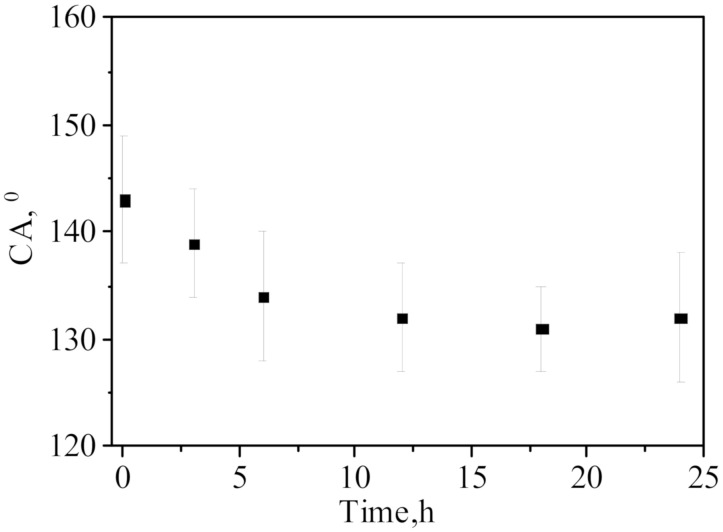
The change in CA of PET-Si-F membranes with pore diameter 152 nm after keeping in water at 75 °C.

**Figure 7 membranes-10-00322-f007:**
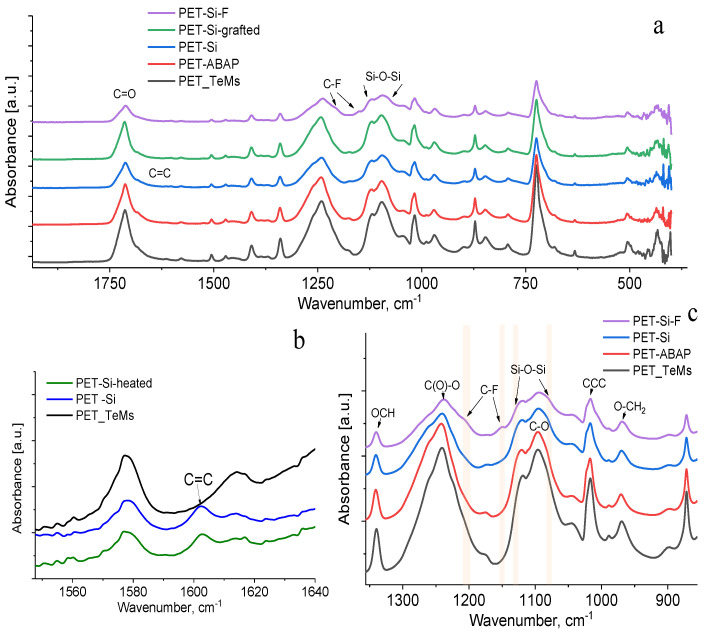
FTIR-ATR spectra of PET ion-track membranes at different stages of modification at wavenumber range from 1800–400 cm^−1^ (**a**) from 1560 to 1640 cm^−1^ (**b**) and from 1350 to 940 cm^−1^ (**c**).

**Figure 8 membranes-10-00322-f008:**
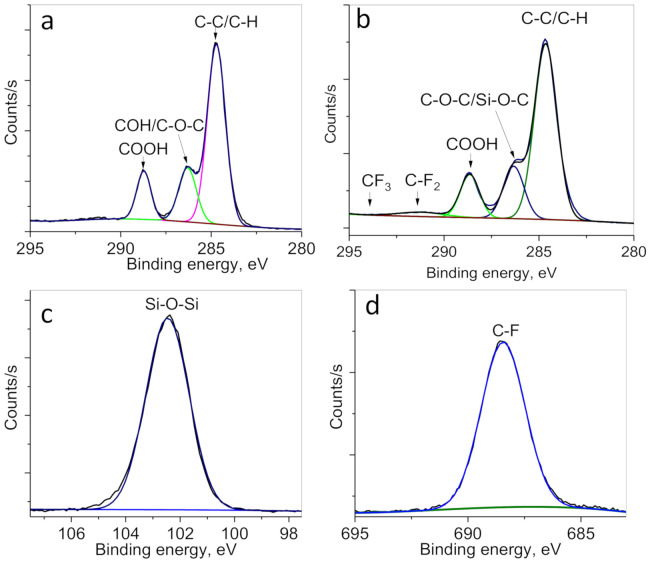
Typical high resolution XPS spectra of C1s (**a**,**b**), Si2p (**c**) and F1s (**d**) of the original PET ion-track membranes (**a**) and PET-Si-F membranes (**b**–**d**).

**Figure 9 membranes-10-00322-f009:**
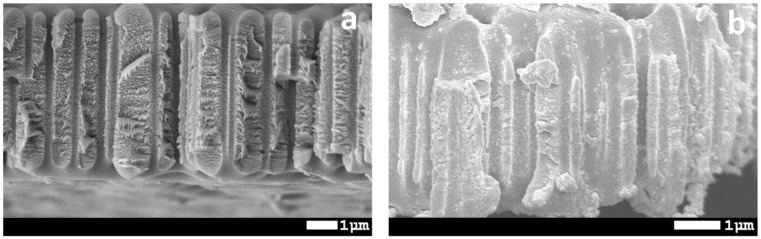
SEM images of cross-sectional view of initial (**a**) and modified (**b**) PET ion-track membranes.

**Figure 10 membranes-10-00322-f010:**
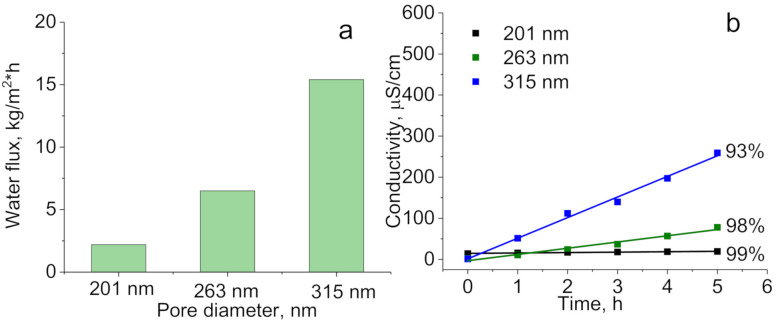
MD water flux (**a**) and electrical conductivity during continuous DCMD tests (**b**) using hydrophobized membranes (PET-Si-F) with different pore sizes (NaCl 30 g/L, temperature 70 °C).

**Figure 11 membranes-10-00322-f011:**
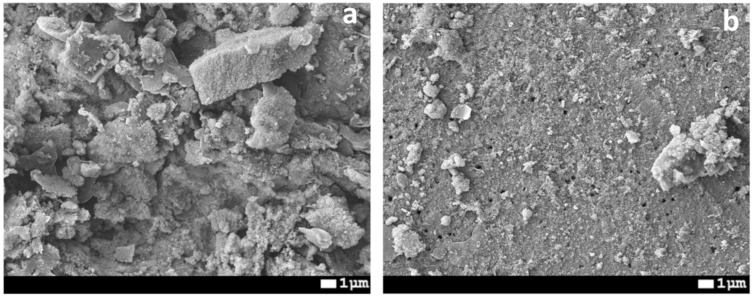
SEM images of PET-Si-F membrane surface (pore diameter is 263 nm) just after DCMD (**a**) and after washing in warm water (**b**).

**Table 1 membranes-10-00322-t001:** CA of PET TeMs with Si NPs obtained at different concentration of surfactant.

Concentration of Surfactant, M	CA *, °
0.0005	117 ± 6
0.0008	114 ± 6
0.0015	109 ± 4
0.002	98 ± 5
0.003	98 ± 5
0.006	71 ± 5

* CA of original PET TeMs is 49 ± 5.

**Table 2 membranes-10-00322-t002:** Contact angle values and pore sizes for grafted/fluorinated PET TeMs at various stages of modification.

Sample	Contact Angle, °	Effective Pore Diameter, nm	LEP_exp_, Bar	LEP_GP-CFD_ Model, Bar
Original PET membrane	49 ± 5	247 ± 5	-	-
PET-ABAP membrane	54 ± 6	247 ± 4	-	-
PET-Si membrane	98 ± 5	167 ± 6	3.2	3.4
PET Si-F membrane	143 ± 6	152 ± 6	> 4.3	9.7
PET Si-F membrane *	135 ± 6	201 ± 5	> 4.3	6.4
PET Si-F membrane **	132 ± 5	263 ± 5	4.1	4.3
PET Si-F membrane ***	125 ± 5	315 ± 6	3.5	3.4

PET membrane with initial pore size of 300 nm *; 350 nm ** and 410 nm ***.

**Table 3 membranes-10-00322-t003:** XPS surface composition of the samples.

Sample	Atomic Concentration, %					High Resolution C1s Moieties, %				
	C	O	N	Si	F	C-C/C-H	Si-O-C/C-O-C	C = O	C-F_2_	C-F_3_
Original PET membranes	72.5	27.5	-	-	-	66.1	19.8	14.1	-	-
PET-ABAP membranes	75.1	22.3	2.6	-	-	66.8	19.4	13.8	-	-
PET- Si membranes	63.9	27.9	0.8	4.1	-	67.1	18.9	14	-	-
PET-Si-F membranes	63.9	27.9	-	5.7	2.5	64.6	19.0	13.5	1.6	0.3

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
