# Peer review of "Modification of PET Ion-Track Membranes by Silica Nanoparticles for Direct Contact Membrane Distillation of Salt Solutions"

_membranes, 2020, doi:10.3390/membranes10110322_

Round 1

Reviewer 1 Report

This paper is concerned with PET ion-track membranes with the silica nanoparticles (Si NPs) immobilized and their applications to direct contact membrane distillation (DCMD) of salt solutions. A proposal about new applications of ion-track membranes would undoubtedly be very interesting for most journal readers because the resulting solid-state nanochannels have a great potential for a future research target. However, in view of the following 16 considerations, the reviewer is sure that the authors will want to carry out some revision of the paper.

On the whole, the reviewer considers the paper a worthwhile contribution to the field of the membrane science and technology, and he believes it deserves to be published.

1. The title should be corrected so that this can represent the content of the paper correctly. The paper covers not only the effect of the Si NPs on the PET ion-track membranes in DCMD, but the sample preparation and characterization. The reviewer notices that the authors’ group published a series of similar studies about the application of the ion-track membranes to DCMD. Please be sure to differentiate this paper from their previous ones.

2. The authors used the pores with diameters of 247, 300, 350, and 410 nm for the preparation of their own samples in this study. The reviewer does not understand the reason for selecting these sizes? Describe what the authors expected for the different pores here.

3. Did you check the geometry of the pores? A pore was rather small and could not always be cylindrical. One concern would be the formation of double conical pores and its effect on the results or DCMD performance in this study. Thus, the authors’ comments and discussion are necessary from a standpoint of the pore shape.

4. Page 3, lines 18-19: “Membranes with Si NPs … of 0.2% ABAP.

The paper says that ABAP was covalently bonded with COOH-end groups of PET in the first stage. If so, why were the membranes immersed in an ethanol solution of ABAP again in the second stage, where the grafting of the Si NPs occurred? The authors need to justify this experimental procedure because, as the authors claim, the novelty of the present paper would lie in the attachment of Si NPs, not the use of the PET ion-track membranes.

5. Page 4: procedure of DCMD experiments

Are the average water flux and degree of salt rejection sufficient enough to evaluate DCMD performance? Discuss briefly the selection of these two parameters in this section. The reviewer also recommends to give a method of estimating the degree of salt rejection.  

6. Page 4, lines 1-2 from the bottom: “SEM analysis was used … average size of NPs

How did the authors use SEM to CONTROL the size of NPs?

7. Table 1:

The data and discussion about the size control of the Si NPs (Table 1, for example) are presented in this paper. Have the similar papers been published so far? If there have really been NO publications about the control of the Si NP size, the authors clearly declare this fact somewhere in this paper. Otherwise, the authors have to introduce other similar studies and explain the significance of the presented data, thereby distinguishing this study from the previous ones.

8. Page 6, last sentence: “Thus, 0.003 M of surfactant … preparation of PET TeMs

The logical flow of the discussion should be reconsidered so that the reviewer can understand the justification of this judgement. Probably, there would be two factors, i.e., the size of the Si NP and CA; are these related to the FTIR and XPS spectra, which exhibited no peaks related to sodium lauryl sulfate? Very confusing.

9. Page 7, lines 7-10: “Vinyl-Si NPs were grafted to … of NPs during exploitation.

The reviewer is interested in the coverage of the Si NPs on the surface and pore wall (see comment 11). Besides, the chemical bonding between the Si NP and PET should be described in Fig. 1.

10. Page 8, lines 1-3: “Analysis of previously published … for PLA fabrics.

The reviewer does not understand what the authors mean here. Do they intend to say that Si NPs were useful to make the membrane surface hydrophobic?

11. General problems about characterization:

Almost all the characterizations using CA measurements, FTIR-ATR, XPS and SEM analyzed the samples from the surface only although the analysis depth was different. No data about the environments in the pores are presented. That would be a serious issue determining the reviewer’s final decision. In his opinion, the density of the COOH-end groups for chemical modification is maybe higher on the pore wall than on the membrane surface because, before the chemical etching, PET is partly degraded at a radius of 100-150 nm in the latent track. Therefore, the environment in each pore would be different from what the authors analyzed in this paper. In reply to this comment, present and discuss persuasively new reasonable data. One possibility can be pursued in “cross-sectional” or “overall” (a whole membrane) analyses and observations.

12. Fig. 9 & Table 4:

The authors just presented the data of XPS surface analysis, but did not discuss anything. Particularly, they should discuss the meaning of the composition change, considering the difficulty in data analysis for carbon and oxygen. If impossible, there would be no need to present Table 4.

13. Fig. 10b:

The reviewer is not sure why the conductivities were above zero and fluctuated at the beginning of experiments, i.e., at time zero. He cannot help but doubt the purity of deionized water, which was said to have a resistance of 18.2 MW.

14. Page 11, lines 13-14: “CA of the membrane … 132±5° to 127±6°.

The CA value is generally determined by the chemical nature and surface roughness. Fig. 11 indicates that both the factors were changed before and after washing in warm water. Continue the discussion about the reason for this finding and what it means.

15. Page 11, lines 15-18: “The results show the … appropriate salt rejection degree.

Conclusions: “The results showed a high … in membrane distillation.

The authors should justify this statement. The main topic is the effect of the Si NPs and, therefore, quantitative comparison in DCMD performance must be made with some previous membranes including the authors’ PET ion-track membranes (without Si NPs) and others (a benchmark membrane if available). Such discussion would enable the readers to understand the potential of the authors’ membranes clearly.

16. The reviewer thinks that the wording and style of many sections in this manuscript need careful editing and would like to suggest, if the authors will submit the revised version, the authors seek the advice of someone with a good knowledge of English, preferably a native writer.

Author Response

Responses are presented in attached file

Reviewer 2 Report

This work report the fabrication and the performance of the track-etched membrane functionalized with hydrophobic nanoparticles for distillation of salt solution. Globally the approach is original and the methodology robust. The paper should be accepted after minor corrections.

In introduction the track-etched membrane should be better introduced and especially the current used and the authors should justify why they choose the track-etched membrane. This recent review should also be mentioned: Ma et al. small methods Doi/10.1002/smtd.202000366

In material and methods section, the reference of chemicals should be mentioned

For the track-etch preparation, UV-light activation is often required prior to the etching. The “certain period” should be replaced by the etching time periods.

L100 the alcohol solution should be clearly detailed

L96 “8” in 108 and “2” should be in superscript.

In the SEM images, the nanoparticle (and pore) diameter are not readable. Please improve the readalibity and provide the number of measured nanoparticles to extract their average size.

Then how are obtained the error bar, number of repetitions.

In the FTIR section, the spectra of raw PET should add to highlight that new peak appear after functionalization. I don’t believe that the peak at 1602 is significant.

A membrane distillation test with the raw membrane should be added to evidence the interest of the functionalization.

Author Response

(The authors gave the same response as above.)
